# On the origin of the extremely different solubilities of polyethers in water

Bernd Ensing [1], Ambuj Tiwari [1,6], Martijn Tros[1,6], Johannes Hunger [2], Sérgio R. Domingos [3], Cristóbal Pérez [3], Gertien Smits [4], Mischa Bonn[2], Daniel Bonn[5] & Sander Woutersen [1]

The solubilities of polyethers are surprisingly counter-intuitive. The best-known example is the difference between polyethylene glycol ($[-CH_2-CH_2-O-]_n$) which is infinitely soluble, and polyoxymethylene ($[-CH_2-O-]_n$) which is completely insoluble in water, exactly the opposite of what one expects from the C/O ratios of these molecules. Similar anomalies exist for oligomeric and cyclic polyethers. To solve this apparent mystery, we use femtosecond vibrational and GHz dielectric spectroscopy with complementary *ab initio* calculations and molecular dynamics simulations. We find that the dynamics of water molecules solvating polyethers is fundamentally different depending on their C/O composition. The *ab initio* calculations and simulations show that this is not because of steric effects (as is commonly believed), but because the partial charge on the O atoms depends on the number of C atoms by which they are separated. Our results thus show that inductive effects can have a major impact on aqueous solubilities.

[1] Van 't Hoff Institute for Molecular Sciences, University of Amsterdam, Science Park 904, 1098XH Amsterdam, The Netherlands. [2] Max Planck Institute for Polymer Research, Department of Molecular spectroscopy, Ackermannweg 10, 55128 Mainz, Germany. [3] Deutsches Elektronen-Synchrotron DESY, Notkestraße 85, 22607 Hamburg, Germany. [4] Swammerdam Institute for Life Sciences, University of Amsterdam, Science Park 904, 1098XH Amsterdam, The Netherlands. [5] Institute of Physics, University of Amsterdam, Science Park 904, 1098XH Amsterdam, The Netherlands. [6]These authors contributed equally: Ambuj Tiwari, Martijn Tros. Correspondence and requests for materials should be addressed to B.E. (email: ensing@uva.nl) or to J.H. (email: hunger@mpip-mainz.mpg.de) or to M.B. (email: bonn@mpip-mainz.mpg.de) or to S.W. (email: s.woutersen@uva.nl)

Polyethers are ubiquitous in daily life and in chemical research. Their solubilities are very different, as is exemplified by the two most common polyethers: PEG (polyethylene glycol, $[-CH_2-CH_2-O-]_n$) dissolves extremely well in water; it is infinitely soluble for $n \leq 600$. It has wide commercial application, and occurs in nearly any cosmetic product. POM (polyoxymethylene, $[-CH_2-O-]_n$) on the other hand, is a plastic (known to every chemist in the form of the Keck clips used to connect glassware) that is completely insoluble in water. The solubilities of these polymers are thus the exact opposite of the textbook prediction based on their C/O ratios[1]. Similar counterintuitive differences in solubility exist for smaller polyethers (Fig. 1): polyethers in which O atoms are separated by two C atoms (red structures in Fig. 1) dissolve much better than their analogs in which the O atoms are separated by one C atom (green structures). The same dichotomy exists for the enthalpies of solution: dissolving a mole of dimethoxyethane (DME, Fig. 1) releases >5 times as much heat as dissolving a mole of dimethoxymethane (DMM)[2,3].

The origin of the different solubilities is not well understood[5]. It is commonly explained by assuming that the distances between the O atoms in different polyethers have consequences for their solubility in water. As early as 1969, Blandamer et al.[6] suggested that water molecules solvating PEG can form a hydrogen-bond network similar to that of bulk water on the basis of the distances between O atoms in the trans-gauche-trans conformation of the OCCO backbone. The resulting good fit of the solvation hydrogen-bond network into that of the surrounding water would then explain the high solubility of PEG. Previous studies[7–22] have shown evidence for this intuitively appealing idea, but to date no systematic experimental investigation of the origin of the different solubilities of polyethers exists. Here, we investigate this issue using spectroscopic experiments in combination with *ab initio* calculations and molecular dynamics simulations. We find evidence that the solubility difference is not due to a difference in hydrogen-bond geometry but has a completely different origin: our results indicate that it is mainly the difference in partial charges on the oxygen atoms that determines the difference in solubility, a result that may be relevant for understanding the solubilities of many other compounds.

## Results

**Time-resolved vibrational spectroscopy**. Figure 1 shows the investigated polyethers. To investigate their solvation, we probe the reorientation of the water molecules in the solutions using time-resolved infrared (IR) pump-probe and dielectric-relaxation spectroscopy, both of which are sensitive probes of water dynamics[23–33]. In the IR experiments we use the OD-stretch vibration as a probe of the reorientation dynamics, in isotopically diluted water (6% HDO:$H_2O$) to avoid the effect of resonant intermolecular coupling[34–38]. The IR-pump pulse excites ("tags") OD groups that are aligned along the IR polarization axis; the resulting anisotropic distribution of vibrationally excited OD

groups is randomized by the random orientational motion of water molecules (a process typically occurring on a picosecond timescale). This causes a decay in the anisotropy parameter $R$, and inversely the anisotropy decay can be used to infer the reorientation dynamics of water molecules. In Fig. 2a we compare the anisotropy decays of solutions of dioxane and trioxane (cyclic PEG and POM oligomers) with that of water. The two polyether solutions have identical molar fractions of ether O-atoms ($x_O = 0.12$). For neat water, the anisotropy decays to zero with a time constant of ~2.5 ps, as reported previously[23]. In the solution of dioxane, a similar decay constant is observed, but there is a residual signal for times >5 ps. This indicates that two types of water OD groups are present: those unaffected by the dioxane, and those whose reorientation dynamics is slowed down because of solvation interactions. The residual anisotropy increases with the amount of added dioxane, which confirms that these slowly reorienting OD bonds solvate the dioxane[29,39–41]. The decaying component of the anisotropy is similar to that of bulk water, which indicates that beyond the solvation shell the OH orientational dynamics is not influenced by the presence of the solute, in agreement with previous findings[33]. Surprisingly, the anisotropy decay of trioxane (cyclic POM oligomer) exhibits a much smaller residual. This difference indicates that water molecules solvating trioxane are much less slowed down by interaction with the solute than are water molecules solvating dioxane. It may be noted that the strong slowdown of water reorientation due to interaction with the solute, as observed in the dioxane solutions, is not necessarily accompanied by a decrease in the OH-stretch frequency: this can happen when the stronger interaction between water and solute (as compared to water–water) is mainly entropic in origin: the solvating OH groups molecules cannot form new hydrogen bonds as easily as the bulk-water OH groups, and this lack of surrounding H-bond acceptors slows down their reorientation (an excluded-volume effect)[42]. A similar situation occurs in solutions of tetramethylurea[43], acetone[44], and DMSO[44], where in each case the solvating OH groups also exhibit slow dynamics, but the OH-stretch frequency is higher than in neat water. For polyether solutions we observe a similar blueshift of the OD-stretch frequency (Fig. 2). We can qualitatively reproduce this blueshift in our density-functional MD simulations (see Supplementary Note 7 and Supplementary Figs. 11–13).

For the linear oligomers, we observe a similar difference in water dynamics (Fig. 2b): again the PEG oligomer exhibits a substantial residual anisotropy (with the same magnitude as the dioxane solution with the same $x_O$), implying a hydration layer with slow water dynamics, whereas the POM oligomer has no residual anisotropy. Solutions of PEG itself (Fig. 2c) show anisotropy decays similar to that of oligomer solutions with the same $x_O$, and are in agreement with recent experiments in which transition–metal complexes were used to probe the water dynamics in PEG solutions. Finally, the response of 1,2-dimethoxypropane (which has an additional methyl group) is the same as that of PEG and 1,2-DME at the same $x_O$, which suggests that the solvating water in the solutions of PEG-like polyethers interacts mostly with the ether O atoms, and less with the methyl groups.

**Dielectric-relaxation spectroscopy**. We complement the infrared experiments with dielectric relaxation spectroscopy (DRS), which probes the polarization of a sample induced by an external oscillating electric field. In the resulting complex permittivity spectra, the real part corresponds to the dielectric permittivity and the imaginary part to the dielectric loss. For neat water, the spectrum is dominated by a dispersion in the real permittivity and a corresponding peak in the dielectric loss at ~20 GHz (Fig. 3,

**Fig. 1** Investigated polyethers and their solubilities[4].

PEG600 ∞   Dioxane ∞   DME ∞   1,2-DMP

POM insoluble   Trioxane 175 g/l   DMM 230 g/l

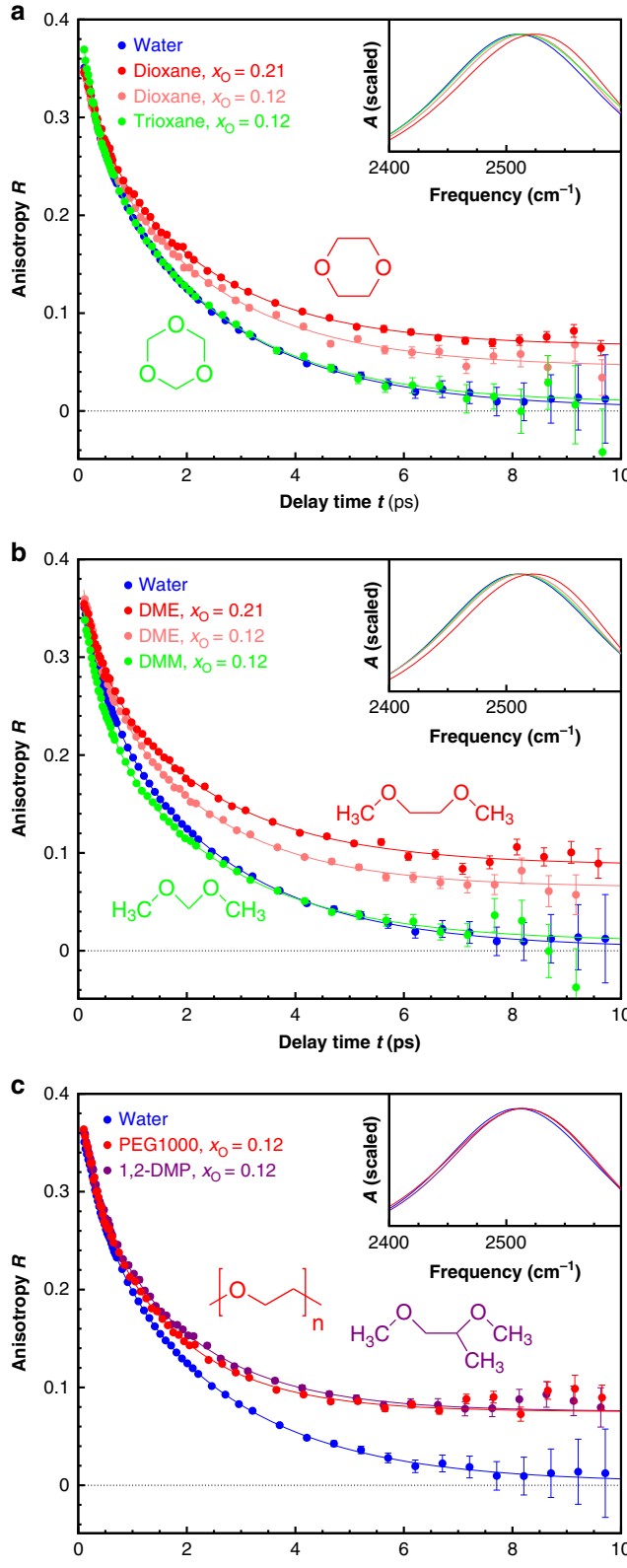

**Fig. 2** OD-Stretch anisotropy decay of different polyether solutions. The solutions are in isotopically diluted water (6% HDO:H$_2$O). The legend indicates the molar fraction $x_O$ of ether O atoms with respect to the total number of molecules in the solution. The curves are least-squares fits (see Supplementary Table 1 for the fit parameters and Supplementary Note 1 for the details of the fit procedure). **a** Cyclic PEG and POM oligomers; **b** the building blocks of PEG and POM; **c** PEG-1000 and 1,2-dimethoxypropane. In each plot, the inset shows the IR absorption spectra of the solutions. The error bars represent $1\sigma$

total electric-dipole moment of all the water molecules, whereas transient IR probes the (second-order) orientational correlation function of spatially well separated OH (or OD) groups (the spatial separation being due to the isotopic dilution)[46]. Rotation of a hydrogen-bond donating water molecule around the hydrogen-bonded OH group does not change the direction of the OH vector (and hence does not contribute to the OH-stretch anisotropy decay), but does rotate the electric-dipole moment (and hence does contribute to the dielectric relaxation); as a consequence, a fraction of solvating water molecules can cause a residual offset in the anisotropy, and an overall slowdown of the dielectric relaxation[47,48]. An additional difference between the two experiments is that DRS probes the collective motion of all the water molecules, and hence is more sensitive to collective water dynamics[47,48]. In Fig. 3 we show the dielectric spectra for solutions of PEG- and POM-like polyethers with increasing $x_O$. The dioxane data are in good agreement with previous studies[49,50]. Besides the reduction in amplitude (mainly due to the reduced volume fraction of water), for both solutes the relaxation shifts to lower frequencies (longer relaxation times). This slow-down of the water dynamics is much more pronounced for DME and dioxane than for DMM and trioxane. To quantitatively analyze this slow-down, we extract the water relaxation times by fitting a Cole–Cole type equation to the spectra[51]. As already indicated by the raw spectra, the relaxation times thus extracted (Fig. 3c, f) increase much more strongly with increasing solute concentration for DME and dioxane than for the DMM and trioxane solutions. Hence, DRS indicates that PEG-like solutes slow down the average water dynamics to a larger extent than POM-like solutes, in agreement with our IR observations. Thus, both the IR-anisotropy and the dielectric-relaxation measurements show that water solvating PEG-like polyethers is much more slowed down by interaction with the solute than water solvating POM-like polyethers.

***Ab initio* and force-field MD simulations**. To elucidate the origin of this difference, we perform *ab initio*, density functional theory molecular dynamics (DFT-MD) and classical forcefield simulations (FF-MD). Due to their broad range of commercial applications, aqueous PEG and short oligomers of polyethylene oxide have been extensively studied with quantum chemical calculations and molecular dynamics simulations[11–13,18,20,22,52–68]. Previous studies have shown that the hydrogen-bond distributions of aqueous DME and PEG are similar, and that insights from studies on short oligomers can help to understand the hydration of longer polymer chains. Various theoretical investigations have focused on the hydration of the different chain conformations of PEG oligomers: the trans-gauche-trans (tgt) conformer being hydrophilic and further stabilized upon hydration, whereas other conformers are more hydrophobic[11,12,18,22,56,63,64]. The translational and rotational diffusion of water slows down monotonically with PEG concentration, while H-bond lifetimes increase[13]. In contrast to PEG-like polyethers, theoretical investigations of POM and methylene-oxide oligomers are scarce. Wada et al. performed free

$x_O=0$). This relaxation results from the random orientational motion of water molecules, with a corresponding relaxation time of ~8.3 ps[45]. It may be noted that the DRS and transient-IR experiments both probe water reorientation dynamics, but the observed relaxation behavior is generally different, because DRS probes the (first-order) orientational correlation function of the

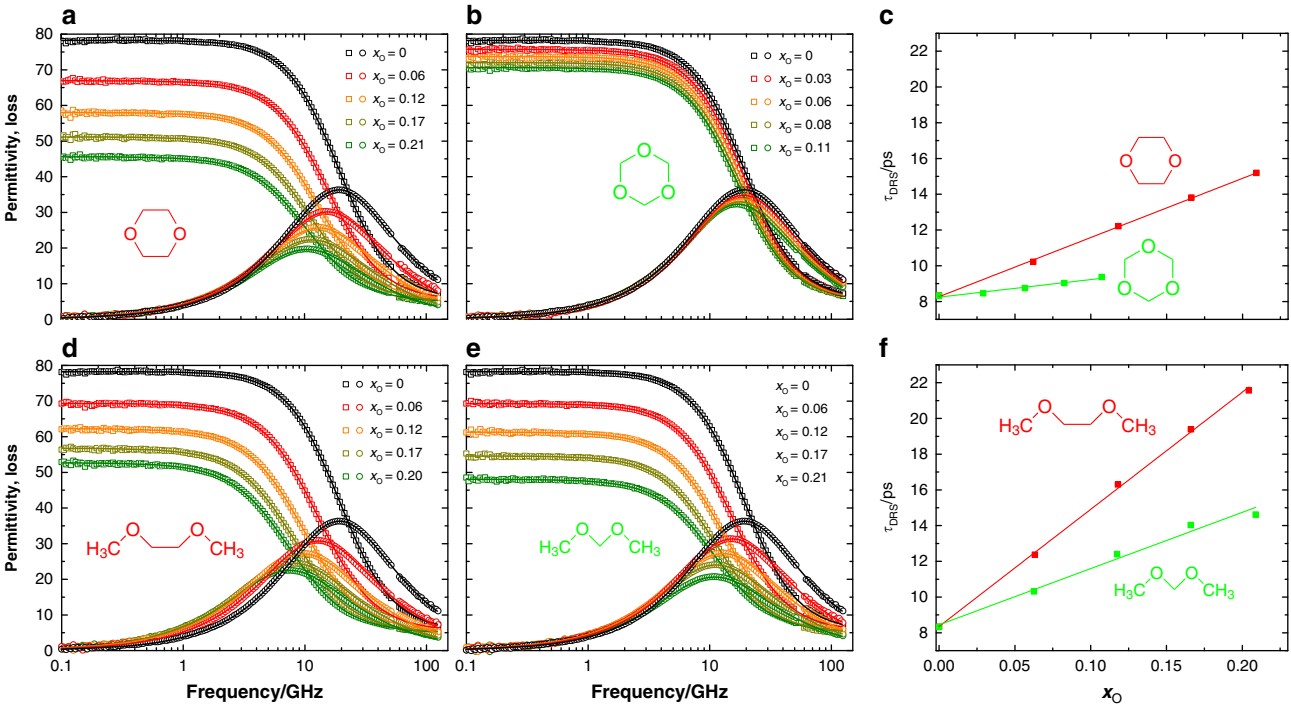

**Fig. 3** Dielectric permittivity (squares) and loss (circles) of polyether solutions. **a** Dioxane, **b** trioxane, **d** DME, and **e** DMM. **c**, **f** show the orientational relaxation times of the water in these solutions, as obtained from least-squares fitting the Cole–Cole equation to the data (fits shown as the curves in **a**, **b**, **d**, **e**

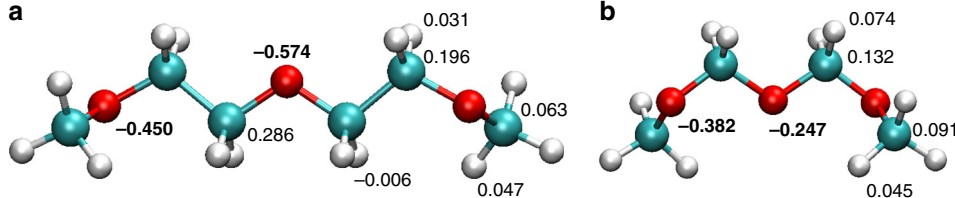

**Fig. 4** Atomic partial charges (RESP) in small polyethers. **a** PEG and **b** POM oligomer. The charges on the O atoms are printed in boldface

energy calculations and report that the tg conformation of DMM only modestly increases its solubility in water, which contrasts with the significant increase of the solubility of DME tgt conformation[22]. This difference provides yet another example of the different solubilities of PEG- and POM-like polyethers.

DFT-MD simulation of DME or DMM in water provides an accurate description of the hydrating solvent structure and dynamics, using atomic forces obtained from the electronic structure and thus including many-body interactions such as polarisation. The computationally less demanding FF-MD simulations, which we benchmark against the DFT-MD results, allow for larger systems and longer timescales[69,70]. We use the Amber GAFF forcefield, in which the atomic (RESP) charges are obtained from fitting the quantum-chemically calculated electrostatic potential around the solute molecule (see Supplementary Note 2 and Supplementary Tables 2 and 3). Apart from DME and DMM, we also perform simulations of oligomers with three ether oxygens, which we refer to as POM3 and PEG3. The atomic charges show remarkable differences between the POM and PEG-like molecules. In particular, the O atoms of the PEG-like molecules have much higher partial charges ($\approx 0.4$–$0.6e$) than those of the POM-like molecules ($\approx 0.2$–$0.4e$), see Fig. 4 and Supplementary Fig. 1. The C and H atoms have smaller, positive charges and are more similar. We thus hypothesize that the

chemical nature of O atoms in POM and PEG-like molecules is different due to an electronic charge distribution that depends on the number of C atoms between the O atoms. This difference in partial O-atom charge can be understood qualitatively in terms of the inductive effect: in POM-like polyethers the electron-withdrawing O atoms must share $CH_2$ groups, whereas in PEG-like polyethers this is not the case. The more negative O atoms in PEG are expected to form stronger hydrogen bonds to the hydrating water solvent than the less negative O atoms in POM.

This expectation is confirmed in Fig. 5, which shows the radial distribution of water H atoms around the central O of a solvated PEG3 or POM3 molecule. PEG3 (black curve) shows a clear hydration peak at $r = 1.8$ Å, which integrates to an average 1.3 water H atoms per PEG3 oxygen. POM3 (red curve) shows hardly any structure; clearly POM3 does not form H-bonds with the solvating water. The DFT-MD simulations confirm the different hydration patterns (see inset). To test if the absence of strong H-bonds to POM3 is due to the less negative oxygens (and not to e.g. steric effects), we perform an additional FF-MD simulation of a fictitious POM3 molecule, in which we set the O charges equal to those in PEG3 (and correspondingly adapt the C charges to maintain a charge-neutral molecule). The radial distribution (green line) indeed shows that the water structure around the modified POM3 is now remarkably similar to that of PEG3,

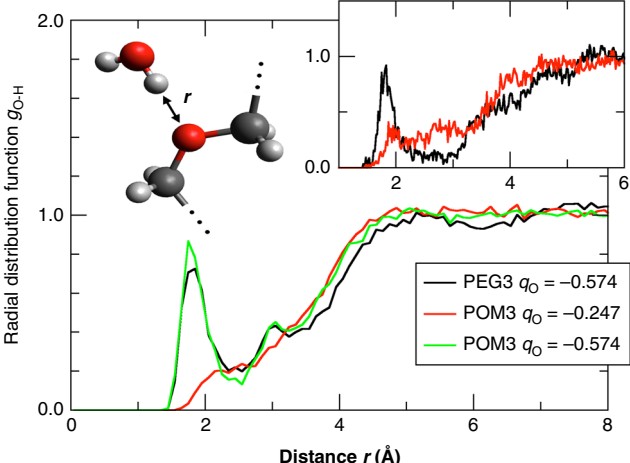

**Fig. 5** FF-MD Radial distribution functions of water H atoms around the solute's middle O atom. For PEG3, the radial distribution function has a hydration peak, which is absent for POM3. Simulation of POM3 with O charges similar to PEG3 recovers the solvent hydration structure (green curve). Inset: DFT-MD results for comparison

confirming that the solute affinity for hydrogen bonding is largely determined by the O charge.

The anisotropy decay was computed for DMM and DME, POM3, and PEG3, and the cyclic trioxane and dioxane molecules. Apart from FF-MD simulations of a single oligomer in water, we also performed simulations at the same $x_O = 0.12$ as in our experimental measurements. Our simulations show that the solvent-water reorientational dynamics slows down significantly in the solutions of PEG-like polyethers, whereas in solutions of POM-like polyethers this is not the case, both in agreement with the experiments (see Supplementary Note 3 and Supplementary Figs. 2–4). By spatially partitioning the solvent, we determine which water molecules cause the observed slowdown in PEG-like solutions. We find a clear slowdown for the waters close to the ether O atoms (distance $r_{H_{water}-O_{solute}} < 2.6\,\text{Å}$). Water molecules near the hydrophobic C atoms ($r_{O_{water}-C_{solute}} < 5.0\,\text{Å}$) also show some rotational slowdown with respect to bulk water, which is due to the excluded volume effect that hydrophobic regions have on the number of hydrogen–bond acceptor and donor sites around a water molecule[42]. At high polyether concentration, the dynamics of the solvating water slows down even more, mainly because hydrating water molecules can interact simultaneously with two solute molecules (see Supplementary Note 3 and Supplementary Fig. 5).

To investigate the difference between the IR-anisotropy and dielectric-relaxation measurements, we also computed the orientational relaxation function of the total electric-dipole moment of the water (which is essentially the Fourier transform of the dielectric spectrum[51,71]) for the PEG3 and POM3 solutions and for bulk water. The results (Supplementary Note 4) show the same qualitative difference as in the experiments: whereas the orientational OH-correlation function of POM3 solution is similar to that of neat water (Supplementary Fig. 2, compare Fig. 2a, b), the dielectric response (Supplementary Fig. 6) slows down significantly compared to neat water, but by much less than in PEG3 solution, in good agreement with the experimental observations (Fig. 3). The difference in O partial charges could also explain the difference in the dynamics of water in solutions of PEG and POM-like polyethers as observed with transient-infrared and dielectric-relaxation spectroscopy. This difference is somewhat more pronounced in the transient-IR than in the

dielectric-relaxation measurements. The more pronounced difference observed in the transient-IR experiments can be understood from the fact that these experiments specifically probe the reorientation of the water OH groups, which can hydrogen-bond directly to the ether O atoms. Hence the observed dynamics will be directly influenced by a difference in ether partial O charges. The dielectric-relaxation experiments on the other hand probe the collective water-dipole relaxation. As such, water dynamics in the hydration shell are correlated to both the dynamics of the solute and to the dynamics in the second hydration shell, which can explain the reduced sensitivity to changes of water dynamics due to varying partial charges.

The rotational slowdown of water molecules hydrating a PEG O atom is thus largely caused by its more negative charge compared to POM. But it is not the only cause, because in that case the O charge should have been at least as negative as a water O atom ($\approx -0.85e$), as was for example previously seen for the zwitter-ionic trimethylamine N-oxide in water[72]. Instead, the mechanism is more akin that of tetramethyl urea[73], where the slowdown at the hydrophilic O atom is enhanced by the neighborhood of hydrophobic groups. These neighboring hydrophobic groups reduce the number of available H-bond acceptors (the excluded volume effect[42]), which enhances the residence time of the hydrating solvent water at the hydrophilic group. In addition to these effects, the small POM-like ethers (which are still soluble in water, see Fig. 1) might exhibit a tendency to form small aggregates. The solutions of these ethers would then not be completely homogeneous on the molecular level, and the correspondingly lower amount of hydrating water could contribute (in addition to the above-mentioned effects) to the observed differences in water dynamics and enthalpy of solution of PEG- and POM-like polyethers, an effect that has been hypothesized to occur for other solutes[74]. To investigate the influence of the ether-O partial charge on the orientational water dynamics, we performed a simulation in which the O charges of a POM3 molecule were changed to those of PEG3 (and the C charges adapted correspondingly to maintain a charge-neutral molecule, see Supplementary Note 2). We find that the conformational distribution of the POM3 molecules does not change significantly upon changing the ether-O partial charges (see Supplementary Note 6 and Supplementary Figs. 9, 10). The resulting anisotropy decay (Supplementary Fig. 4) is very similar to that of PEG3 solution, indicating that the different ether-O partial charges in PEG and POM-like polyethers are the main cause of the observed difference in orientational water dynamics in solutions of these polyethers.

## Discussion

Our experimental and computational results suggest a new explanation for the different solubilities of polyethers: water interacts more strongly with PEG-like polyethers than with POM-like polyethers (Fig. 5) as a consequence of the larger partial charge on the O atoms in the former (Fig. 4). The larger partial charges of PEG-like polyethers would also result in a larger enthalpy of hydration as compared to POM-like polyethers, and this could partly explain the larger negative enthalpy of solution[2,3], and hence the better solubility. It should be noted that in general, solubility is determined not only by the hydration strength. Dissolving a substance can be regarded energetically as a process involving two steps, each of which is accompanied by an enthalpy change: (1) removing the molecules (or ions, in the case of a salt) from their pure solid or liquid phase, and (2) subsequent hydration of these free molecules (or ions). The net enthalpy change when dissolving a substance in water is the sum of these two contributions. In the case of polyethers, the enthalpy change of the

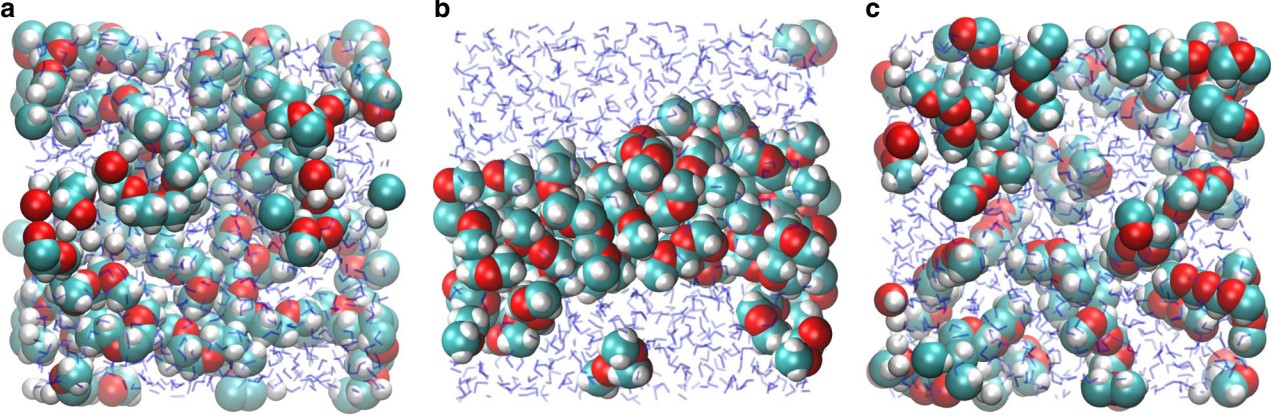

**Fig. 6** Snapshots after 1 ns of FF-MD simulation. **a** PEG3, **b** POM3, and **c** fictitious POM3 molecules with modified atomic charges. Water molecules are indicated in blue. Time series of snapshots are provided in Supplementary Fig. 8, together with a clustering analysis obtained from a 50 ns simulation

first step (removing a molecule from its pure liquid phase) is very similar for PEG- and POM-like polyethers of similar size: for $CH_3OCH_2CH_2OCH_3$ and $CH_3OCH_2OCH_3$ the vaporization enthalpies at room temperature are $+36.6$ and $+31.2$ kJ mol$^{-1}$ respectively[75,76], a difference of only $+5.4$ kJ mol$^{-1}$. On the other hand, the heats $\Delta H_{sol}$. released upon dissolving these substances are $-59.1$ and $-10.5$ kJ mol$^{-1}$ for $CH_3OCH_2CH_2OCH_3$ and $CH_3OCH_2OCH_3$ respectively[3], a difference of $-48.9$ kJ mol$^{-1}$. Using Hess' law we can therefore conclude that the hydration enthalpies of these two ethers differ $-54.3$ kJ mol$^{-1}$ in favor of the PEG-like polyether. The difference in polyether solubility is thus predominantly due to the hydration interaction, i.e. to the difference in hydration enthalpies (the much smaller difference in the enthalpies required to remove the ethers from their pure liquid state actually works in the opposite direction); and the simulations indicate that this stronger hydration interaction is mostly due to the higher partial O charges (as observed in the simulations). The anchoring of the hydrating water molecules due to the presence of the neighboring hydrophobic groups (the excluded-volume effect[42]) probably further enhances the solubility.

As already mentioned, it is commonly believed that the different solubilities of POM and PEG are caused by the different distances between their O atoms. To confirm that the charge of the O atoms rather than such steric effects explains the different solubilities we performed a simple *in silico* experiment (see Supplementary Note 5 and Supplementary Figs. 7, 8 for details). We find that starting from fully mixed configurations, PEG3 molecules remain well dissolved in water (Fig. 6a), whereas POM3 precipitates during the simulation by forming a single large aggregate (Fig. 6b). If we now start from this phase-separated configuration and only change the O charges to the values they have in PEG (and correspondingly adapt the C charges to maintain a charge-neutral molecule), then within 500 ps the POM3 precipitate spontaneously dissolves, confirming unambiguously that the charge distribution in the molecule is responsible for its solubility (Fig. 6c).

To conclude, our results suggest that the very different solubilities of polyethers are mainly caused by inductive effects, resulting in the different partial charges of the O atoms in these molecules. This difference in partial charges, enhanced by the anchoring of the water molecules hydrating the neighboring hydrophobic ether groups[42], can also explain the experimentally observed difference in water dynamics. We believe that similar inductive effects may explain many other instances of counter-intuitive solubilities; and that it is essential to take these effects into account when predicting the solubilities of new compounds.

## Methods

**Sample preparation.** Ether solutions were prepared by mixing the appropriate amount of ether with 97% of the total water volume. The resulting solution was divided over two containers; to one of these 3% $D_2O$ was added, to the other 3% $H_2O$. In the latter, H/D exchange results in solutions with 6%HDO in the water. The basic ether:water molar ratio chosen was 15:1, based on the solubility of dimethoxymethylene (33%). PEG1000 is a polymer with mass 1000 g/mol, consisting of 22 monomers and therefore has 22 oxygen atoms. As DME contains two oxygen atoms for PEG1000 a concentration of 11 times lower was used (ratio 165:1). Additionally, for DME, 1,4-Dioxane, 1,3,5-Trioxane and PEG1000 more concentrated solutions with water:ether molar ratio 7.5:1 for the first three and 82.5:1 for the latter were measured.

**Infrared spectroscopy.** Fourier-transform infrared (FTIR) measurements were done by a Perkin–Elmer Spectrum Two spectrometer with a resolution of 0.5 cm$^{-1}$. The OD-stretch spectra of the ether:HDO:$H_2O$ solutions were corrected for the (small) background absorption by subtracting spectra of solutions with the same ether:water ratio but prepared with pure $H_2O$.

**IR pump-probe measurements.** The IR sample cell for the pump-probe measurements consisted of two 1 mm-thick CaF$_2$ windows separated by a 25 $\mu$m teflon spacer. Polarization-resolved TRIR spectroscopy experiments were done as follows[48]. The output of a commercial Ti:sapphire laser (Coherent Legend Elite, pulse energy 3 mJ) was used to pump a commercial OPA (Coherent OPerA Solo). The output of the OPA was used to generate 20 $\mu$J mid-infrared light (bandwidth ~200 cm$^{-1}$ FWHM) with difference-frequency generation in AgGaS$_2$. Probe and reference pulses were obtained from the mid-IR by reflection off the front and back surfaces of a wedged BaF$_2$ window; the remainder was used as the pump pulse. The polarization of the pump pulse was set at 45° with respect to that of the probe pulse using a MgF$_2$ zero-order $\lambda$/2 plate. The polarization of the measured probe spectrum was selected using a polarizer placed after the sample and set at either 0 or 90° with respect to the pump polarization. The absorption change induced by the pump pulse was monitored by the mid-IR probe pulse, which is spatially overlapped with the pump beam, at various time delays between the pump and probe pulses. The mid-IR pump, probe and reference beams were focused through the sample by means of an $f = 100$ mm off-axis parabolic mirror. At the sample, the probe mid-IR beam diameter was ~200 $\mu$m. Transient absorption changes were measured by frequency-dispersed detection of the mid-IR pulses using a $2 \times 32$ HgCdTe (MCT) array detector (Infrared Associates). Pulse-to-pulse fluctuations in the IR intensity were corrected for using the spectrum of the reference pulse, which was measured simultaneously with that of the probe pulse. To obtain accurate anisotropy data, the thermal contribution to the pump-probe signal was subtracted using the procedure of Rezus and Bakker[43].

**Dielectric relaxation spectroscopy.** We determine complex permittivity spectra $\varepsilon(v) = \varepsilon'(v) + i\varepsilon''(v)$ of these samples as a function of field frequency $v$ with a frequency domain reflectometer at $0.1 \le v/\mathrm{GHz} \le 125$ by combining data obtained using two different experimental geometries: we recorded permittivity spectra at $0.1 \le v/\mathrm{GHz} \le 51$ with a coaxial reflectometer based on an Anritsu Vector Star MS4647A vector network analyser with an open ended coaxial probe[77], based on 1.85 mm coaxial connectors[78]. Spectra at $60 \le v/\mathrm{GHz} \le 125$ were covered using an open ended coaxial probe, based on 1 mm coaxial connectors, connected to an external frequency converter module (Anritsu 3744 A mmW module)[79]. To calibrate the instrument for directivity, frequency response, source match errors we use water[45], air, and conductive silver paste as calibration standards[77]. The spectra thus

obtained are shown in Fig. 3. To obtain the dielectric relaxation times (Fig. 3c, f), we fit a Cole–Cole relaxation model to the experimental spectra[51].

**Quantum-chemical calculations and simulations**. Born-Oppenheimer DFT-MD simulations were carried out with the combined Gaussian and plane-wave (GPW) method as implemented in the the CP2K software, using the BLYP + D3 exchange-correlation functional, norm-conserving pseudopotentials to describe the nuclei and core electrons, and a DZVP Gaussian basis set, combined with a 280 Ry planewave expansion, to describe the valence electrons. The classical MD simulations were carried out with the LAMMPS program, using a fully flexible Amber GAFF forcefield and the fully flexible TIP3P water model. Further details are found in Supplementary Note 2.

## Data availability
The data that support the findings of this study are available from the corresponding authors on reasonable request.

## Code availability
All codes written for and used in this study are available from the corresponding authors upon reasonable request.

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

## Acknowledgements
We would like to thank Michiel Hilbers and Hans Sanders for technical support, and Huib Bakker for stimulating discussions. This research is part of the MaxWater initiative of the Max-Planck Society.

## Author contributions
D.B., G.S., B.E., and S.W. conceived the experiments; M.T. prepared the samples and carried out the experiments; J.H. and M.B. supervised the dielectric–relaxation experiments and data analysis; S.W. supervised the vibrational-spectroscopy experiments and data analysis; A.T. carried out the simulations and their analysis; B.E. supervised the simulations and their analysis; S.R.D. and C.P. helped in the interpretation. All authors contributed to writing the manuscript.

## Additional information

**Competing interests:** The authors declare no competing interests.

