## [Peer Review File · Nature Communications]

Reviewers' comments:

Reviewer #1 (Remarks to the Author):

The manuscript by Ensing et al. reports a combined experimental and computational study aimed to explain the origin of the different solubility of polyether in water. The problem addressed is interesting and the experimental data seems shown quite distinctly the differences in the hydration behavior of the investigated poly-ethers. However, on this part, I cannot deeply evaluate the quality of the results since it is not my main area of expertise.

I have instead several concerns on the modeling and computational part of the manuscript listed in the following points:

1) I was expecting to find the full detail of the simulation methods (in particular the FF-MD ones) if not in the manuscript, at the least in the supporting information but, indeed, they are completely omitted. For example:

a. What is the composition of the simulated systems in term of box size, number of molecules, the length of the simulations?

b. What model has been used for the water model (I guess the TIP3 but it needs to be indicated!).

Such information together with the protocol adopted to perform the simulations (e.g. cutoffs, time steps, electrostatics, thermostat, pressure control algorithms, etc.) should be reported to allow the reproduction of their computational experiment.

2) The authors mentioned that they have studied DME and DMM and chains of longer lengths but it is not explained why results are reported only for PEG3 and POM3 simulations. Why the RDF's for the other system not reported?

3) As above, the benchmarking the FF-MD model against the DFT-MD is an interesting idea but only some results (RDF) for one system are shown.

4) The "in silico" experiment is a nice idea but how the partial charges have been distributed on the other atoms of the POM3 system?

5) It is not indicated at which simulation time the snapshots of Fig.6 have been extracted from the simulation trajectory. A snapshot of the simulation does not tell much about the average distribution of the molecules in the system along with a long MD simulation. Probably RDF's calculated with respect the PEG and POM carbon atoms of the different chains would give you a more quantitative idea of the contacts among the distribution of the chains over longer simulation times (at the least 50-100 ns).

6) In the literature, there are different high-quality computational models and QM studies of PEG, DME, DMP, PEO, PPO. Some of these studies (not all of them and the authors are invited to look more carefully in the literature) have been cited in the introduction. However, the comparison and discussion of the proposed models and experimental results with the previous literature models is completely omitted.

Reviewer #2 (Remarks to the Author):

This work employs femtosecond vibrational and GHz dielectric spectroscopy with complementary ab initio calculations and molecule dynamics simulation to address the counter-intuitive solubilities of polyethers. The authors claimed the partial charge on the O atoms depends on the number of C atoms by which they are separated is responsible for solubility difference. This topic is very important in understanding the aqueous solubility.

There are a few comments on the content of this manuscript. The manuscript should be major revised on these points before it is accepted for publication.

1. The experimental results show the water solvating in the solutions of PEG-like polyethers is stronger than that of POM-like polyethers. The experimental data is quite good, but I am not sure that the ability of solute solvating water is related with the solubility or not. In my opinion, the strong solvation solute do not related with high solubility, and some weak solvation solute may have quite high solubility, for example strong hydrated ions and weakly hydrated ions. So what's the correlation between the solubility and solvation ability in polyether solutions?

2. For Fig.6, you mentioned you only changed the O charges to the value they have in PEG. Is this means the POM changes to charged particle? If this is so, then the segregation of POMs is only because the repulsion of charged particles.

Reviewer #3 (Remarks to the Author):

This manuscript concerns the interesting solvation behavior of polyethers in water. It is known that molecules that have the ether oxygens spaced by two carbons (PEG-like) are exceedingly soluble, while molecules with the ether oxygens spaced by a single carbon (POM-like) are virtually insoluble. This presents a puzzle since one would naively have guessed the higher oxygen content would promote solubility.

The present manuscript presents results from transient IR and dielectric relaxation measurements combined with theoretical *ab initio* calculations and MD simulations that aim to explain this puzzling phenomena.

The manuscript presents some very interesting results and a compelling argument for the charge on the oxygens playing the major role, thus making it potentially appropriate for *Nature Communications*. However, in the present form the manuscript does not present a consistent picture from experiments and theory, which in my view limits the broader impact of the study. Some of the details of the manuscript need to be explained further and the arguments need to be sharpened to provide an overall consistent picture.

The experiments show that the water dynamics is slower for PEG-like molecules attributed to a stronger interaction between the molecule and water thus promoting the solubility, while POM-like molecules do not affect the water dynamics. This is explained by differences in the charges on the oxygens calculated in the *ab initio* calculations: "The more negative O atoms in PEG are expected to form stronger hydrogen bonds to the hydrating water solvent than the less negative O atoms in POM." However, such stronger hydrogen-bonds would lead to a red-shift in the OH spectrum, where the experimental IR spectra, as presented in this article, show a blue-shift.

Furthermore, the transient IR measurements show that the timescale for the water orientation itself does not change but rather that PEG induces a static offset. This would imply that the water dynamics is just as fast around PEG, but that the available orientational space is confined (the water molecules can not rotate across all angles) within a 10 ps time frame. [A minor point here is that the fit parameters in the SI are given in terms of the rates and not timescales, which make it a bit harder to compare the actual fits to the described timescales in the main manuscript.]

This also seems in contrast to the dielectric relaxation measurement, which shows a continuous change in the dielectric relaxation time, with PEG increasing the timescale about twice as much as POM. It would be good to connect the two measurements, beyond simply that both slow down dynamics. In particular, it would be informative to quantitatively and specifically relate the experimental observations with the theoretical investigations to present an overall picture of the water dynamics. In this regard, the calculated radial distribution functions for the POM3 with $q=-$

0.574 is quite telling in terms of the water structure. Perhaps simulating and calculating the anisotropy decay for the same molecule would be informative on the relative effects of the partial charge and the excluded volume on the water dynamics.

**Reply to the reviews of manuscript NCOMMS-17-13917-T
“The origin of the extremely different solubilities of polyethers in water”**

We kindly thank all three reviewers for their constructive comments on our manuscript. In the revision of the text, we have taken all the reviewers' comments and suggestions into account, and we believe this has resulted in a more convincing and clearer manuscript. We have also carried out the additional MD simulations suggested by reviewers #1 and #3. Performing and analyzing these additional simulations, which corroborate our earlier conclusions, has caused some delay in the resubmission, for which we apologize.

Our answers to the reviewers' comments are given below, in the same order as they appear in their reviews. In the PDFs of the revised manuscript and Supplementary Information, all modified and added text is highlighted. In our reply, the reviewer comments are printed in italics and our answers in normal font.

Reviewer #1

1. I was expecting to find the full detail of the simulation methods (in particular the FF-MD ones) if not in the manuscript, at the least in the supporting information but, indeed, they are completely omitted. For example:

a. What is the composition of the simulated systems in term of box size, number of molecules, the length of the simulations?

b. What model has been used for the water model (I guess the TIP3 but it needs to be indicated!). Such information together with the protocol adopted to perform the simulations (e.g. cutoffs, time steps, electrostatics, thermostat, pressure control algorithms, etc.) should be reported to allow the reproduction of their computational experiment.

AUTHOR REPLY We fully agree with the referee and apologise for not providing a full description of the computational details before. The supporting information now contains a new section, in which all relevant computational details of the *ab initio* DFT-MD simulations and the forcefield MD simulations are compiled, including the system composition, box size, simulation lengths, thermostat and barostat settings, forcefield details, electrostatics, cut-offs, etc. We indeed use the (fully flexible version of the) TIP3P forcefield for the water description. In the main article, we have expanded the section on the computational details in the “Method” section, and we have added a reference to the SI for the further details.

CHANGE TO TEXT Main article: We have expanded the paragraph on the computational details in the “Method” section and we refer to the SI for the further details SI: A new section “MD SIMULATIONS” has been added, in which all computational details are compiled.

2. *The authors mentioned that they have studied DME and DMM and chains of longer lengths but it is not explained why results are reported only for PEG3 and POM3 simulations. Why the RDFs for the other system not reported?*

AUTHOR REPLY We apologize for the confusion. The only longer oligomers that we investigated in addition to DME and DMM (“PEG2” and “POM2”) were PEG3 and POM3. Hence, our defining a notation PEG x and POM x was somewhat misleading, as it suggests that a whole range of x values was investigated. This was not the case: the short oligomers ($x = 2, 3$) already provided sufficient information for the present investigation, in which we focused on the difference in CO- and CCO-type polyether solubility. In the main text, we focus on the PEG3 and POM3 solutions, because this allows us to distinguish between the central oxygen atom and the two side oxygen atoms (see for example Table SIII in the SI) in our analysis. In the SI we report the results for all four investigated systems.

CHANGE TO TEXT

Modified sentence defining PEG3 and POM3 in the main text. Added the results on the O-H stretch anisotropy from the FF-MD simulations on the DME and DMM solutions in Figure S2 and table SIII in the Supporting Information.

3. *As above, the benchmarking the FF-MD model against the DFT-MD is an interesting idea but only some results (RDF) for one system are shown.*

AUTHOR REPLY The Amber Gaff forcefield and the water TIP3P forcefield are well-established for the study of hydrophilic, hydrophobic and amphiphilic solvation. The additional DFT-MD simulations were carried out as an extra test for the FF-MD simulations and also for analysis of the electronic structure of the solutes (we were wondering if we could see the differences in the POM and PEG oxygen charges from the Mulliken charges or from the maximally localized Wannier orbital center positions during the dynamics in full water solvent; unfortunately, Mulliken charges exhibit too large fluctuations, whereas the Wannier center positions are too insensitive). For the benchmarking, we indeed compared the radial distribution of solute atoms with respect to water (as shown in Figure 5). For dynamical properties, in particular comparison of anisotropy decay, the DFT-MD simulations were not long enough for a statistically meaningful verification.

4. *The in silico experiment is a nice idea the but how the partial charges have been distributed on the other atoms of the POM3 system?*

AUTHOR REPLY In addition to the note in the text “we perform an additional FF-MD simulation of a fictitious POM3 molecule, in which we set the O charges equal to those in PEG3 (and adapted the C charges to maintain a charge-neutral solute molecule)”, we have now added a figure to the SI (Figure S1), in which the charges on the atoms are explicitly shown.

CHANGE TO TEXT Addition of Figure S1 in the SI, showing the PEG3, POM3,

and modified POM3, molecular structures with the atomic charges.

5. It is not indicated at which simulation time the snapshots of Fig.6 have been extracted from the simulation trajectory. A snapshot of the simulation does not tell much about the average distribution of the molecules in the system along with a long MD simulation. Probably RDFs calculated with respect the PEG and POM carbon atoms of the different chains would give you a more quantitative idea of the contacts among the distribution of the chains over longer simulation times (at the least 50-100 ns).

AUTHOR REPLY We agree with the reviewer. Indeed, the snapshots shown in Figure 6 only serve to illustrate the description in the main text of our observations on the solvation and desolvation behavior of the polyether oligomers in the simulations. To supply further proof of our observations, we now provide a time series of snapshots from all three simulated systems in the Supplementary information together with a cluster analysis.

CHANGE TO TEXT We now mention in the caption of Figure 6 that the snapshots are after 1 ns of simulation time and we refer to the Supplementary Information of a time lapse over the full 50 ns of simulation for each system. Figure S8 in the Supplementary Information shows now snapshots after $t = 0.01, 0.3, 0.5, 20,$ and 50 ns simulation time. Figure S7 in the Supplementary Information quantifies these observations for each system with graph of the largest cluster size, obtained from a cluster analysis along the 50 ns simulations. A description of both Figures is provided in the Supplementary Information.

6. In the literature, there are different high-quality computational models and QM studies of PEG, DME, DMP, PEO, PPO. Some of these studies (not all of them and the authors are invited to look more carefully in the literature) have been cited in the introduction. However, the comparison and discussion of the proposed models and experimental results with the previous literature models is completely omitted.

AUTHOR REPLY We thank the reviewer for his/her suggestion, which have taken to heart in revising the manuscript. We have added a new paragraph (on page 3) in which we discuss the previous computational work on polyethers. In particular, we discuss the work on the different solubilities of PEG (trans/gauss) conformers and on the solvent water mobility and hydrogen-bond network. We also discuss an interesting earlier paper in which the different solubilities of DMM and DME were investigated [Wada, et al., J. Phys. Chem. B 118, 12223 (2014)].

CHANGE TO TEXT Added references to the introduction; added a new paragraph on page 3 in which we cite the previous computational work done on polyether systems.

Reviewer #2

1. The experimental results show the water solvating in the solutions of PEG-like polyethers is stronger than that of POM-like polyethers. The experimental data is quite good, but I am not sure that the ability of solute solvating water is related with the solubility or not. In my opinion, the strong solvation solute do not related with high solubility, and some weak solvation solute may have quite high solubility, for example strong hydrated ions and weakly hydrated ions. So whats the correlation between the solubility and solvation ability in polyether solutions?

AUTHOR REPLY We thank the reviewer for bringing up this important point, which we have addressed in the revised manuscript. The strength of the solvation interaction is indeed not the only determinant of solubility, and as the reviewer remarks, some ions with weak solvation interaction may dissolve just as well in water as other ions with strong solvation interaction. As an illustration, one can think of LiF vs. LiCl: F^- has much stronger solvation interaction than Cl^- (due to its smaller ionic radius), but nevertheless LiF is less soluble in water than LiCl. The reason is that dissolving involves two steps: (1) removing the ions (or molecules) from their pure solid (or liquid) phase, and (2) subsequent solvation of these free ions (or molecules). It follows from Hess' law that the net enthalpy change when dissolving a substance in water is the sum of these two contributions. In particular, in the case of LiF vs. LiCl we have for the total enthalpy of dissolution*

$$\begin{aligned}\Delta H_{\text{solution}}(\text{LiCl}) &= \Delta H_{\text{LiCl} \rightarrow \text{Li}^+(\text{g}) + \text{Cl}^-(\text{g})} + \Delta H_{\text{Li}^+(\text{g}) \rightarrow \text{Li}^+(\text{aq})} + \Delta H_{\text{Cl}^-(\text{g}) \rightarrow \text{Cl}^-(\text{aq})} \\ &= +840 - 519 - 515 = -244 \text{ kJ mol}^{-1}, \\ \Delta H_{\text{solution}}(\text{LiF}) &= \Delta H_{\text{LiF} \rightarrow \text{Li}^+(\text{g}) + \text{F}^-(\text{g})} + \Delta H_{\text{Li}^+(\text{g}) \rightarrow \text{Li}^+(\text{aq})} + \Delta H_{\text{F}^-(\text{g}) \rightarrow \text{F}^-(\text{aq})} \\ &= +1034 - 519 - 515 = 0 \text{ kJ mol}^{-1},\end{aligned}$$

which explains why LiF dissolves so much worse than LiCl.

In the case of the polyethers, the exact same energetic considerations apply (with smaller energies because the interactions are weaker). *The important point is that the energy of the first step (removing a molecule from its pure liquid phase) is very similar for PEG- and POM-like polyethers of similar size.* For $\text{CH}_3\text{OCH}_2\text{CH}_2\text{OCH}_3$ and $\text{CH}_3\text{OCH}_2\text{OCH}_3$ the vaporization enthalpies at room temperature[†] are +36.6 and +31.2 kJ mol^{-1} respectively, a difference of only +5.4 kJ mol^{-1} . On the other hand, the heats $\Delta H_{\text{solution}}$ released upon dissolving these substances[‡] are -59.1 and -10.5 kJ mol^{-1} for $\text{CH}_3\text{OCH}_2\text{CH}_2\text{OCH}_3$ and

*The numbers are experimental data from D.W. Smith, *J. Chem. Educ.* **54**, 540 (1977) and C.R. Gopikrishnan *et al.*, *AIP Advances* **2**, 012131 (2012).

[†]The values are experimental data from Steele *et al.*, *J. Chem. Eng. Data* **41**, 1285 (1996) and J.S. Chickos *et al.*, *J. Phys. Chem. Ref. Data* **32**, 519 (2003).

[‡]The values are experimental data from Barannikov *et al.*, *J. Chem. Thermodynamics* **43** 1928 (2011).

CH₃OCH₂OCH₃ respectively, a difference of $-48.9 \text{ kJ mol}^{-1}$. Using Hess' law we can therefore that the solvation enthalpies of these ethers differ $-54.3 \text{ kJ mol}^{-1}$ in favor of the PEG-like polyether; and the difference in polyether solubility is thus completely due to the solvation interaction, i.e. to the difference in solvation enthalpies (the much smaller difference in the enthalpies required to remove the ethers from their pure liquid state even works in the opposite direction).

CHANGE TO TEXT We explain this point on p. 4 of the revised manuscript.

2. For Fig.6, you mentioned you only changed the O charges to the value they have in PEG. Is this means the POM changes to charged particle? If this is so, then the segregation of POMs is only because the repulsion of charged particles.

AUTHOR REPLY The total charge of the modified POM molecule was kept neutral by adapting accordingly the charges of the central carbon atoms. In the revised text this is stated whenever appropriate to avoid confusion (on p .4 and twice on p. 5). We have added a figure to the SI (Figure S1), in which the atomic charges are explicitly shown.

Reviewer #3

1. The experiments show that the water dynamics is slower for PEG-like molecules attributed to a stronger interaction between the molecule and water thus promoting the solubility, while POM-like molecules does not affect the water dynamics. This is explained by difference in the charges on the oxygens calculated in the ab initio calculations: The more negative O atoms in PEG are expected to form stronger hydrogen bonds to the hydrating water solvent than the less negative O atoms in POM. However, such stronger hydrogen-bonds would lead to a red-shift in the OH spectrum, where the experimental IR spectra, as presented in this article, show a blue-shift.

AUTHOR REPLY As the reviewer points out, stronger hydrogen bonds can indeed cause a red-shift in the OH spectrum, but this happens only in the case that the H-bond strength increase is predominantly energetic in origin; that is, it would occur if the oxygen partial charge of the solute were more negative than that of solvent water molecules (see for example the cases of tetramethylurea, acetone, and DMSO, where in each case the solvating OH groups also exhibit slow dynamics, but the OH-stretch frequency is nevertheless higher than in neat water, as reported in Refs. [48] and [49] of the revised manuscript). However, in the present case the origin of the stronger binding of water to the solute than to the solvent is mainly entropic. The more negative oxygen charge in PEG increases the H-bond affinity substantially compared to POM hydration, but not more than that of water. The entropic effect due to embedding of the PEG oxygens inbetween the hydrophobic aliphatic atoms, which reduces the probability for hydrating solvent molecules

to find a new H-bond binding partner and leave the solute coordination shell (the well-known excluded volume effect), causes an addition effective H-bond strength increase, but no red-shift in the OH-stretch spectrum.

CHANGE TO TEXT In the revised manuscript, we have added the above explanation on p. 2 (right column).

2. Furthermore, the transient IR measurements show that the timescale for the water orientation itself does not change but rather that PEG induce a static offset. This would imply that the water dynamics is just as fast around PEG, but that the available orientational space is confined (the water molecules can not rotation across all angles) within a 10 ps time frame. [A minor point here is that the fit parameters in the SI are given in terms of the rates and not timescales, which make it a bit harder to compare the actual fits to the described timescales in the main manuscript.]

AUTHOR REPLY We thank the reviewer for raising this issue. A residual offset in the OH-stretch anisotropy $R(t)$ can be due to (i) a subensemble of OH-groups coordinated to the solute, and exhibiting extremely slow reorientation; this results in a bi-exponential decay of $R(t)$, with the second time constant so slow that it appears as a residual offset in the experimentally accessible time range; or (ii) re-orientation of the OH groups in a restricted solid angle (this is the mechanism that the reviewer mentions). Previous time-resolved IR studies [Fenn et al., *JACS* **131**, 5530 (2009); Daley et al., *J. Phys. Chem. B* **121**, 10574 (2017)] have shown that in polyether solutions, scenario (i) applies. We have clarified this issue in the revised manuscript.

CHANGE TO TEXT We explain the origin of the residual offset in the anisotropy in the main text, with references to the above-mentioned studies.

3. This also seems in contrast to the dielectric relaxation measurement, which show a continuous change in the dielectric relaxation time, with PEG increasing the timescale about twice as much as POM. It would be good to connect the two measurements, beyond simply that both slow dynamics. In particular, it would be informative to quantitatively and specifically relate the experimental observations with the theoretical investigations to present an overall picture of the water dynamics. In this regard, the calculated radial distribution functions for the POM3 with $q=-0.574$ is quite telling in terms of the water structure. Perhaps simulating and calculating the anisotropy decay for the same molecule would be informative on the relative effects of the partial charge and the excluded volume on the water dynamics.

AUTHOR REPLY We thank the referee for these remarks and suggestions. There are indeed interesting qualitative differences between the IR and dielectric-relaxation results. As the reviewer remarks, the IR-anisotropy decay shows a

residual offset for the solutions of PEG-like polyethers, whereas the dielectric-relaxation shows an increase of the time constant. Such differences between IR-anisotropy and dielectric-relaxation measurements have been observed previously for aqueous solutions and are well understood, see e.g. *Science* **328**, 1006 (2010), *J. Chem. Phys.* **137**, 044503 (2012) and *J. Chem. Phys.* **141**, 18C535 (2014). The IR-anisotropy and dielectric-relaxation experiments can give very different results because they probe the reorientation of different axes of the water molecules, *viz.* the OH groups of water (HDO) molecules and the electric-dipole moment of the water molecules, respectively. These OH-bond and dipole-moment vectors are at an angle, and the breaking and making of hydrogen bonds can have very different effects on their respective dynamics: in particular, rotating a water molecule around a hydrogen-bond donating OH group leaves the OH vector unchanged, but rotates the electric-dipole moment. Hence, strongly hydrogen-bonded OH groups can cause a residual offset in the IR-anisotropy and a slowdown (rather than a residual offset) in the dielectric relaxation function. Another difference between the two types of measurement is that the HDO molecules probed in the IR-anisotropy measurements are well separated (the experiments are done on HDO in isotope-diluted water), so that these measurements probe the dynamics of uncorrelated water molecules, whereas the dielectric-relaxation is sensitive to all water molecules, and therefore more sensitive to correlations in the reorientational motions of neighboring water molecules. Such correlations are very strong in water (and characterized by the so-called Kirkwood factor, see the *J.Chem.Phys.* articles referred to above). Thus, the two types of measurements generally cannot be compared directly.

We have taken to heart the reviewer's suggestion to perform additional MD simulations to shed more light on these issues. We focus on what we consider the most surprising difference between the IR-anisotropy and dielectric-relaxation measurements, namely that in solutions of POM-like polyethers the IR-anisotropy decay is essentially the same as that of neat water (Fig. 2a,b), whereas the dielectric relaxation shows an increase of the relaxation time constant (Fig. 3), albeit a significantly smaller increase than in PEG-type solutions. As the reviewer suggested, we have performed additional simulations to investigate this difference in more detail. The simulations reproduce the experiments very well: comparing the orientational correlation functions of the OH-groups (probed in the IR anisotropy) and of the total electric-dipole moment (probed in the dielectric relaxation) obtained from the simulation, we find the same qualitative difference as in the experiments.

We have also carried out the reviewer's second suggestion to simulate the anisotropy decay of the same polyether molecule with PEG-like and POM-like charges. The results are shown on p. 3 of the Supporting Information. The results show that the partial charge is the main cause of the orientational slowdown observed in solutions of PEG-like polyethers. After modifying the oxygen charge

of POM3 to that of PEG3, the solvent water molecules around the central POM3 oxygen show a similar rotational slowdown as in the PEG3 system (see Fig. 2 of the Supporting Information). The minor discrepancy is probably due to the smaller excluded-volume effect. We thank the reviewer for suggesting this idea to us.

CHANGE TO TEXT In the revised text, we explain more extensively than we did before why in general the IR-anisotropy and dielectric-relaxation measurements can give different results. We have added a section to the Supporting Information describing the simulations of the correlation function of the electric-dipole moment, and we have added a brief discussion in the main text in which we present and discuss these additional findings. Finally, we have added the details of the simulated anisotropy decays of POM3 with PEG charges in the Supporting Information, and we discuss the results of these simulations in the main text.

Reviewers' comments:

Reviewer #1 (Remarks to the Author):

The authors have addressed all my criticisms with satisfactory answers. I don't have further comments and in my opinion, the manuscript is now suitable to be published in the Nature Comm.

Reviewer #2 (Remarks to the Author):

I read through the revised manuscript, all of my questions have been seriously addressed. I have no more questions. I recommend it is accepted for publication.

Reviewer #3 (Remarks to the Author):

The manuscript present interesting results concerning a big fundamental chemical problem that is interesting for a broad audience and thus relevant for Nature Chemistry.

The manuscript has been improved in several ways in response to the reviews, including the additional MD simulations I suggested.

However, the language is still somewhat confusing and the arguments and the overall take-home message need to be sharpened before publication.

The main result is that the big discrepancy in the solvation is due to inductive effects, i.e. the charge on the oxygen atoms rather than steric effects, as previously suggested.

While the MD simulations clearly show that when the charges on the atoms are changed artificially, the properties of the molecules can be reversed strongly supporting this idea, the experiments tell a more nuanced story.

As the authors convincingly argue in their response to my review, the IR experiments indicate that the slower dynamics are due to entropic effects, i.e. restriction of the number of hydrogen-bonding partners, and not enthalpic (which would have led to a red-shift and not a blue-shift in the vibrational frequency). I realize that this perhaps is differences in semantics, but I would be careful calling this that the water is more strongly bound to the PEG-like molecules, as it does in the current manuscript, as this in my opinion imply an enthalpic effect.

The use of the term “strong” for the binding of the water in this case is confusing, which is further propagated in the discussion on page 5 of the manuscript: “The stronger binding of water to PEG-like polyethers results in a much larger enthalpy of hydration as compared to POMlike polyethers, and this explains the larger negative enthalpy of solution,^{2,3} and hence the better solubility. It should be noted that in general, solubility is determined not only by the hydration strength.”

Which is it, is the difference in the water-molecule interactions between the PEG and POM like molecules mainly enthalpic or entropic in nature?

Also, if the difference in solvation is driven by induction, but the strength of the hydrogen-bonding (i.e. spectral shift or the enthalpy) is not changed significantly, there must be a structural change between the two systems, as also evidenced by the radial distribution function in Figure 5. The increased negative charge in the PEG-like systems cause a structural change that include a subpopulation of restricted water molecules. This is also collaborated by the comment in the manuscript that the change in charge on the oxygen cannot fully explain the slower water dynamics but must be collaborated by the hydrophobic groups. It cannot be that the structure of POM and PEG-like molecules are the same except that some strongly bound water molecules are missing. Do the MD simulations show indications of a change in structure between the PEG and POM-like molecules?

Finally, I acknowledge that the transient IR and dielectric relaxation experiments measure different quantities, but it is still surprising that the transient IR show not effect for POM like molecules compared to water and a change for PEG like molecules, indicating a qualitative difference or the appearance of restricted water molecules; but that the dielectric relaxation experiments show a gradual change that is just smaller for POM vs PEG like molecules. It would seem that this would tell us something about the differences between the systems.

In short, the main conclusion that the difference in solvation is due to induction seems only a part of the story, where the experiments provide insight into more nuanced effects. The language needs to clearly separate entropic and enthalpic effects and the conclusion needs to be sharpened.

Reply to the reviews of revised manuscript NCOMMS-17-13917-T
“The origin of the extremely different solubilities of polyethers in water”

We were happy that the first two reviewers were content with our revisions, and we have carefully followed the additional suggestions of reviewer #3. Below we list reviewer #3’s additional comments (in italics) and our corresponding revisions:

- *The language is still somewhat confusing and the arguments and the overall take-home message need to be sharpened before publication. While the MD simulations clearly show that when the charges on the atoms are changed artificially, the properties of the molecules can be reversed strongly supporting this idea, the experiments tell a more nuanced story.*

We have formulated the conclusions more precisely, and we have tried to make our title less absolute and some of the sentences in the introduction and conclusion more nuanced.

- *I realize that this perhaps is differences in semantics, but I would be careful calling this that the water is more strongly bound to the PEG-like molecules, as it does in the current manuscript, as this in my opinion imply an enthalpic effect.*

We completely agree with the reviewer’s suggestion not to use the expression “strongly bound” for water molecules that have slow orientational dynamics: we have rephrased the text wherever appropriate and now refer to “slowdown of water reorientation” instead of “strongly bound water”.

- *Which is it, is the difference in the water-molecule interactions between the PEG and POM like molecules mainly enthalpic or entropic in nature?*

The reviewer’s question is difficult to answer from our experiments and simulations; probably the interaction contains an enthalpic and entropic component (the difference in hydration enthalpies indicates that the enthalpic component is not insignificant). For this reason we would prefer not to speculate on this issue.

- *It cannot be that the structure of POM and PEG-like molecules are the same except that some strongly bound water molecules are missing. Do the MD simulations show indications of a change in structure between the PEG and POM-like molecules?*

We addressed the reviewer’s question if there is a change in structure between PEG and POM-like molecules by performing additional simulations. We find that the conformation of the polyethers does not change significantly upon changing the partial charges from PEG- to POM-like. We state this explicitly in the revised text, and the simulation results are included in the SI.

- *Finally, I acknowledge that the transient IR and dielectric relaxation experiments measure different quantities, but it is still surprising that the transient IR show not effect for POM like molecules compared to water and a change for PEG like molecules, indicating a qualitative difference or the appearance of restricted water molecules; but that the dielectric relaxation experiments show a gradual change that is just smaller for POM vs PEG like molecules. It would seem that this would tell us something about the differences between the systems.*

As proposed by the reviewer, we have discussed the difference between the IR and dielectric measurements in more detail (on p. 5 of the revised manuscript).

In the submitted PDFs of the manuscript and supplementary information, all changes have been highlighted. We thank the third reviewer for his/her comments and suggestions, which we believe have improved and clarified our argumentation, and we hope that this version will be suitable for publication in Nature Communications.

REVIEWERS' COMMENTS:

Reviewer #3 (Remarks to the Author):

I am satisfied with the authors' corrections to the manuscript and will now recommend it for publication as is.